# Policy-shaped prediction: avoiding distractions in model-based reinforcement learning

**Miles Hutson**
Stanford University
hutson@stanford.edu

**Isaac Kauvar**
Stanford University
ikauvar@stanford.edu

**Nick Haber**
Stanford University
nhaber@stanford.edu

## Abstract

Model-based reinforcement learning (MBRL) is a promising route to sample-efficient policy optimization. However, a known vulnerability of reconstruction-based MBRL consists of scenarios in which detailed aspects of the world are highly predictable, but irrelevant to learning a good policy. Such scenarios can lead the model to exhaust its capacity on meaningless content, at the cost of neglecting important environment dynamics. While existing approaches attempt to solve this problem, we highlight its continuing impact on leading MBRL methods —including DreamerV3 and DreamerPro— with a novel environment where background distractions are intricate, predictable, and useless for planning future actions. To address this challenge we develop a method for focusing the capacity of the world model through synergy of a pretrained segmentation model, a task-aware reconstruction loss, and adversarial learning. Our method outperforms a variety of other approaches designed to reduce the impact of distractors, and is an advance towards robust model-based reinforcement learning.

## 1 Introduction

Model-based reinforcement learning (MBRL) is a promising path to data-efficient policy learning, and recent advances show impressive performance with high dimensional sensory data [Hafner et al., 2023]. A central component of MBRL is a world model, which is trained to predict how an agent's actions impact future world states. However, the world is highly complex while the capacity of a world model is finite, and ultimately only a subset of the components and dynamics of the environment can be accurately modeled. In this setting, distracting stimuli can be particularly problematic, as they waste the capacity of the world model on useless details.

To address the challenge of distractors, a number of MBRL methods seek to isolate the most important components of an environment, including structural regularizations [Deng et al., 2022, Fu et al., 2021, Wang et al., 2022], pretraining the agent's visual encoder [Seo et al., 2022, Wu et al., 2023], and value-equivalent world modeling [Schrittwieser et al., 2020], while environments such as Distracting Control Suite have been developed to assess distractor suppression [Stone et al., 2021].

In this paper we introduce a new method, Policy-Shaped Prediction (PSP), for identifying and focusing on the important parts of an image-based environment. Rather than relying on pre-imposed structural regularizations, PSP learns to prioritize information that is important to the policy. We synergize task-informed gradient-based loss weighting, use of a pre-trained segmentation model [Kirillov et al., 2023] and adversarial learning to create a distraction-suppressing agent that outperforms leading image-based MBRL agents. In addition to exhibiting similar performance in distraction-free settings and on a standard benchmark of robustness to distractions, our method markedly improves performance in the face of particularly challenging distractors that are intricate but entirely learnable. Because learnable distractors can be accurately modeled, they straightforwardly contribute to reducing the world model's error, but needlessly exhaust the capacity of the world model.

38th Conference on Neural Information Processing Systems (NeurIPS 2024).

In sum, we make the following key contributions:

- We describe Policy-Shaped Prediction (PSP), a MBRL method that achieves strong distraction suppression by combining gradient-based loss weighting with a pretrained segmentation model to focus learning on important environment features.
- We augment PSP with a biologically-inspired action prediction head that reduces sensitivity to self-linked distractions.
- We introduce a challenging new benchmark for testing robustness to learnable distractions.
- We demonstrate that PSP achieves 2x improvement in robustness against challenging distractions while maintaining similar performance in non-distracting settings.

## 2    Policy-Shaped Prediction

We introduce PSP, a method to reduce an agent's sensitivity to useless distractions by focusing on sensory stimuli that are most relevant to its policy, rather than seeking to model everything in the environment. Our guiding intuition was that we can use the gradient from the policy to the input image to identify important pixels in the environment, and that we can aggregate these pixelwise salience signals to identify important objects by using image segmentation. Specifically, we extend the principles of VaGraM [Voelcker et al., 2022] to a high-dimensional vision model by using explainability-related notions of salience and aggregating an otherwise noisy gradient-based salience signal within objects. Additionally, inspired by the biological concept of efference copies [Crapse and Sommer, 2008], which are neural signals used to cancel out sensory consequences of an animal's actions, we incorporated a way to explicitly mitigate distractions caused by actions of the agent itself.

PSP employs (1) gradients of the policy with respect to image inputs to identify task-relevant elements of the image, (2) a segmentation model to aggregate gradients within each object in the image, and (3) an adversarial objective to the image encoder of the world model that discourages encoding of duplicate information about the previous action. Figure 1 illustrates the training modifications made by this method to the underlying DreamerV3 [Hafner et al., 2023] architecture. Notably, since these modifications only affect the training stage of the world model, the DreamerV3 agent remains unaltered during inference. Below, we describe each of the three key components in detail.

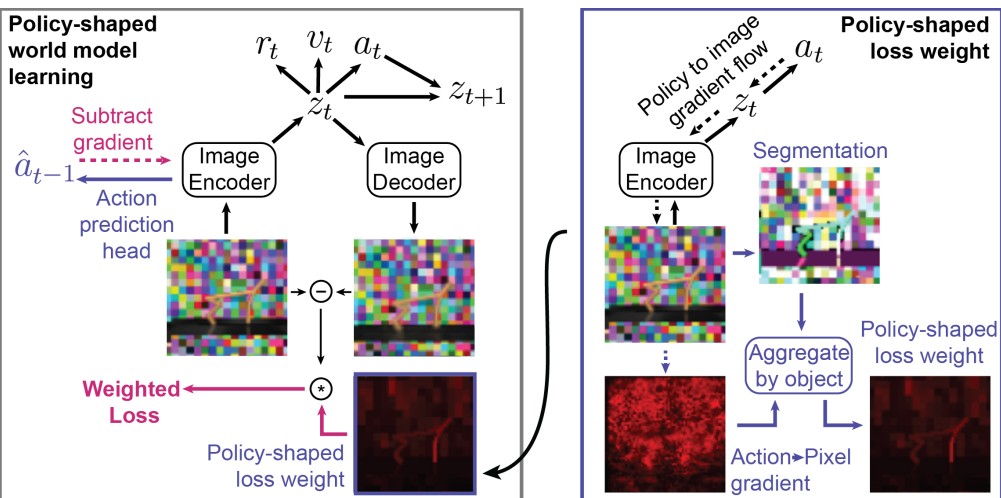

Figure 1: Policy-Shaped Prediction in an environment with challenging distractions. (left) Training of an otherwise-unaltered DreamerV3 agent is modified in two ways: 1) A head is added to predict the previous action based on the image encoding, and the gradient of the head is subtracted from the gradient of the image encoder, and 2) the loss is scaled pixelwise by a policy-shaped loss weight. (right) The loss weight uses the gradient of the policy to the input pixels. The image is segmented, and the pixel weights are averaged within each segmented object. Dashed lines signify gradient flow.

## 2.1 Task-informed image reconstruction with policy-gradient weighting

Our approach builds upon the core idea that signals most important to the actor and/or critic should be given special importance in the world model. The concept of using the critic to inform model loss was applied in Value-Gradient weighted Model loss (VaGraM) [Voelcker et al., 2022], which weights the model loss according to the gradient of the value function with respect to the state. We extend this concept to high-dimensional image inputs, which previous work did not demonstrate. This extension to the image domain is inspired by gradient-based interpretability methods such as saliency maps [Simonyan et al., 2013, Shrikumar et al., 2017, Ancona et al., 2019].

By upweighting the reconstruction loss for parts of the image that inform value estimation, we might expect to improve the performance of a downstream policy that aims to maximize value. Going one step further, we propose using the gradient of the policy for weighting the model loss. While VaGraM focused solely on the value function, we hypothesize that the gradient of the policy may provide an even more informative signal – because ultimately, the state representation must support effective action selection. We hypothesize that the set of signals informing action selection may be richer than those that inform value estimation, which might rely primarily on simple cues such as whether an agent has flipped over. In contrast, the signals needed to select actions can be more subtle, such as the distance of an agent's leg from the platform it pushes off of in order to run.

To compute the policy-gradient weighting, we first sum across the dimensions of the action vector $\mathbf{a} = \mathbb{E}(\pi(\mathbf{s}))$, where $\mathbf{s}$ is the latent state of the world model, to produce a scalar $a = \sum_j a_j$, and then take the gradient with respect to the pixels of the input image $x$. To apply this weighting in the context of DreamerV3 [Hafner et al., 2023], we scale the image reconstruction loss term at each pixel $i$, for reconstructed image $\hat{x}$.

$$\mathcal{L}_{\text{image}}(\phi) = \sum_{x_i} \frac{\partial a}{\partial x_i}(\hat{x}_i - x_i)^2 \tag{1}$$

## 2.2 Object-based aggregation of gradient weights

Gradient-based weighting of the world model's reconstruction ultimately is a form of applying model explainability methods, which attempt to highlight the most important elements of a model's input for its outputs. From model explainability literature, a known challenge with gradient-based weighting is its noisiness, which is likely caused by the presence of sharp but meaningless fluctuations in the derivative at small scales [Smilkov et al., 2017]. While this has been combated by more computationally demanding explainability approaches such as Integrated Gradients [Sundararajan et al., 2017] and SmoothGrad [Smilkov et al., 2017], we found these to be infeasible to run within the train loop, since this would require taking the derivative of the function with respect to multiple varying inputs for every example in the original input batch. Instead, to combat this problem we introduce a second novel contribution: object-based aggregation of an explainability signal using a segmentation model (SEG). In principle many models should work, but to ensure we had high quality segmentations, we used the Segment Anything Model (SAM) [Kirillov et al., 2023], a pre-trained and broadly applicable segmentation model. Nothing prevents a different model from being utilized, so long as it is of sufficient quality. This idea follows from the observation that the saliency map tends to show a broad, if noisy, correlation with relevant regions in the image (e.g. Figure 1). During data collection, we segment each image into object masks using the existing method laid out by the authors of SAM. We prompt the model with a grid of 256 points, filter the resulting masks with metrics for the SAM algorithm (Intersection-Over-Union and a "stability score"), followed by non-maximum suppression. We also include a mask to capture pixels that are not otherwise assigned to an object. Then, instead of weighting the image reconstruction loss with the raw gradient-based salience, we use a salience score that is aggregated within each segmentation mask. The weight of a pixel $x_i$ that has been assigned to segment $\text{SEG}(x_i)$ is the mean absolute value weight of the pixels in $\text{SEG}(x_i)$.

$$W_i = \frac{1}{||\text{SEG}(x_i)||} \sum_{j \in \text{SEG}(x_i)} |\partial a / \partial x_j| \tag{2}$$

To ignore any exploding gradients, we clip the raw salience map to the 99th percentile before aggregation. Gradients are sometimes near zero in the beginning of training, and in the rare case that all gradients are zero, we set $W_i = 1$ for all $i$, As a regularizer, we linearly interpolate between the salience weighting and a uniform weighting, with $\alpha = 0.9$ for all our experiments, and using rescaled $W_i' = \text{width} \cdot \text{height} \cdot W_i / \sum_i W_i$ to match the scale of the uniform background.

$$W_i'' = \alpha W_i' + (1 - \alpha) \tag{3}$$

This regularizer lets the world model maintain reasonable reconstruction of less-salient aspects of the environment, which the actor-critic function can use as it learns. Without some degree of reconstruction of less-salient regions, the model can become trapped in local minima with a bad policy and world model. Also, at the start of training, gradients from actor-critic functions are essentially random and often small. A uniform background offers a reasonable prior to begin the train loop.

### 2.3 Adversarial action prediction head

We additionally sought to explicitly reduce the distracting sensory impact of an agent's own actions. As animals move, they experience sensory signals generated by their actions and the external environment, and they have evolved the ability to distinguish these signals using efference copies [Crapse and Sommer, 2008]. We hypothesized that we could separate information about an agent's actions and its encoding of external stimuli through domain-adversarial training [Ganin et al., 2016]. To this end, we introduce an adversarial action prediction head that prevents the model from wasting capacity on irrelevant stimuli that are created by the agent's own actions.

The DreamerV3 world model consists of three main components: a convolutional neural network (CNN) image encoder $z_t \sim q_\phi(z_t|h_t, e_t)$ with $e_t = \text{CNN}_\rho(x_t)$, which processes the input image, serves as a prior during training, and encodes the environment state during inference; a recurrent state space machine (RSSM) consisting of $h_t = f_\phi(h_{t-1}, z_{t-1}, a_{t-1})$ and $\hat{z}_t \sim p_\phi(\hat{z}_t|h_t)$ that is trained to simulate the progression of latent states given actions; and an image decoder, $\hat{x}_t \sim p_\phi(\hat{x}_t|h_t, z_t)$ which reconstructs the image from the latent state. Problematically, the encoder can capture information about previous actions from the image, despite this information already being provided directly to the RSSM through the action input. In other words, $z_t$ may source information about $a_{t-1}$ directly through $x_t$, despite $a_{t-1}$ being an argument to $f_\phi$ during the computation of $h_t$. Unfortunately, our reconstruction loss weighting may not solve this problem, since during backpropagation from the actor-critic functions, we do not distinguish information about previous actions that comes from the image versus the action input to the RSSM.

To prevent the CNN encoder from wasting capacity on encoding duplicate information about an agent's actions, we add a small multilayer perceptron (MLP) head that is optimized to predict the previous action from the image embedding.

$$\hat{a}_{t-1} = \text{MLP}_\omega(\text{stop\_grad}(\text{CNN}_\rho(x_t))) \tag{4}$$

$$\mathcal{L}_{\text{AdvHead}}(\hat{a}_{t-1}, a_{t-1}) = (\hat{a}_{t-1} - a_{t-1})^2 \tag{5}$$

When updating $\theta$ during world model training, we *subtract* the scaled gradient $\epsilon \cdot \nabla_\theta \mathcal{L}(\hat{a}_{t-1}, a_{t-1})$ from the overall world model gradient, with $\epsilon = 1e3$. This forces the latent state's previous action information to come solely from the provided action vector.

Our training procedure for a DreamerV3 agent is shown in Algorithm 1. We note that it should be possible to apply these concepts of gradient-based weighting, segmentation-based aggregation, and adversarial action prediction to world models other than our chosen DreamerV3 architecture.

## 3 Experiments

To evaluate the model's performance we design our experiments around the following questions:

Q1. Is our agent robust against distractors which are learnable by the world model, but of no utility for the actor-critic?

Q2. What aspects of the environment are assigned importance by our method?

Q3. Is our agent robust against distractors that are unrelated to the agent's actions?

Q4. Does our agent maintain performance in standard, lower-distraction environments?

Q5. What are the contributions of each component of our method?

---

**Algorithm 1** Policy-Shaped Prediction training (for DreamerV3)

---

1: **Input:** World model parameterized by $\phi$, policy $\pi$ parameterized by $\theta$, image encoder parametrized by $\rho$, replay buffer with image transitions $(x_{t-1}, a_{t-1}, x_t, r_t, c_t)$, SEG segmentation model (SAM, in our application), action prediction MLP parameterized by $\omega$

2: **for** training iteration $1, 2, \dots$ **do**

3:    Sample batch of transition sequences

4:    $G = \nabla_x \pi_\theta$               # Gradient of policy with respect to input image pixels

5:    $S = \text{SEG}(x)$                      # Segmentation of input image

6:    $W = \text{agg}(G, S)$                  # Aggregate gradient using segmentation

7:    $W_i' = \text{width} \cdot \text{height} \cdot W_i / \sum_i W_i$        # Normalize weighting

8:    $W'' = \alpha W' + (1 - \alpha)\mathbf{1}_{\text{shape}}(W')$      # Linearly interpolate with uniform weighting

9:    $\mathcal{L}_{\text{pred}}(\phi) = -\ln p_\phi(x_t \mid z_t, h_t) \odot W'' - \ln p_\phi(r_t \mid z_t, h_t) - \ln p_\phi(c_t \mid z_t, h_t)$
                                         # Weighted DreamerV3 prediction loss

10:   $\mathcal{L}(\phi) = \mathbb{E}_{q_\phi} \left[ \sum_{t=1}^{T} (\beta_{\text{pred}}\mathcal{L}_{\text{pred}}(\phi) + \beta_{\text{dyn}}\mathcal{L}_{\text{dyn}}(\phi) + \beta_{\text{rep}}\mathcal{L}_{\text{rep}}(\phi)) \right]$ # DreamerV3 model loss

11:   $\hat{a}_{t-1} = \text{MLP}_\omega(\text{stop\_grad}(\text{CNN}_\rho(x_t)))$       # Adversarial action prediction head

12:   $\phi \leftarrow \text{Adam}(\nabla\mathcal{L} - \epsilon * \partial\mathcal{L}(\hat{a}_{t-1}, a_{t-1})/\partial\rho, \phi)$

13:   $\mathcal{L}_{\text{AdvHead}}(\hat{a}_{t-1}, a_{t-1}) = (\hat{a}_{t-1} - a_{t-1})^2$

14:   $\omega \leftarrow \text{Adam}(\nabla\mathcal{L}_{\text{AdvHead}}, \omega)$

15: **end for**

---

## 3.1 Experimental details

**Baselines** We test four Model-Based RL approaches as baselines: DreamerV3 [Hafner et al., 2023], and three methods specifically designed to handle distractions – Task Informed Abstractions [Fu et al., 2021], Denoised MDP (method in their Figure 2b) [Wang et al., 2022], and DreamerPro [Deng et al., 2022]. Additionally, we choose DrQv2 [Yarats et al., 2021a] as a representative baseline Model-Free approach. For all agents, we use 3 random seeds per task, and default hyperparameters.

**Environment details** Visual observations are $64 \times 64 \times 3$ pixel renderings. We test performance in three environments: DeepMind Control Suite (DMC) [Tassa et al., 2018], Reafferent DMC (described below), and Distracting Control Suite [Stone et al., 2021] (with background video initialized to a random frame each episode, 2,000 grayscale frames from the "driving car" Kinetics dataset [Kay et al., 2017]). For each environment, we test two tasks: Cheetah Run and Hopper Stand. We selected these tasks because they present different levels of difficulty, allowing us to assess how distraction-sensitivity depends on task difficulty. For ablation experiments, we test on Cheetah Run.

## 3.2 Reafferent Deepmind Control Suite

In the natural world, distractions can be highly complex, but in many cases are also highly predictable. For instance, the creaking sound a rusty bicycle makes as you pedal, or the movement of your own shadow as you dance outside. We wanted an environment which would allow investigation of how well existing methods would perform in scenarios where the representational complexity of the distractions are very high, but they cannot simply be ignored as 'unlearnable' noise. Distinguishing this type of partly self-generated distraction requires identifying which parts of the world are relevant to taking action, not just those affected and unaffected by our action. To achieve this, we devised the Reafferent Deepmind Control environment, in which the distracting background images have substantial content, but they depend deterministically on

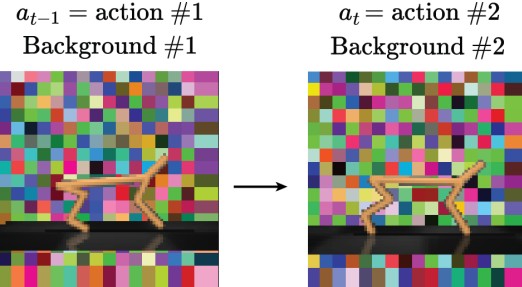

$a_{t-1} = $ action #1          $a_t = $ action #2
Background #1             Background #2

Distracting background is complex, but *entirely predictable* based on agent's previous action.

Figure 2: Schematic of the Reafferent Deepmind Control environment. The distracting background is entirely predictable based on the agent's previous action and the elapsed time in the episode.

the agent's previous action and the elapsed time in the episode – and are thus completely predictable (Figure 2). We build on the Distracting Control Suite [Stone et al., 2021], using a background

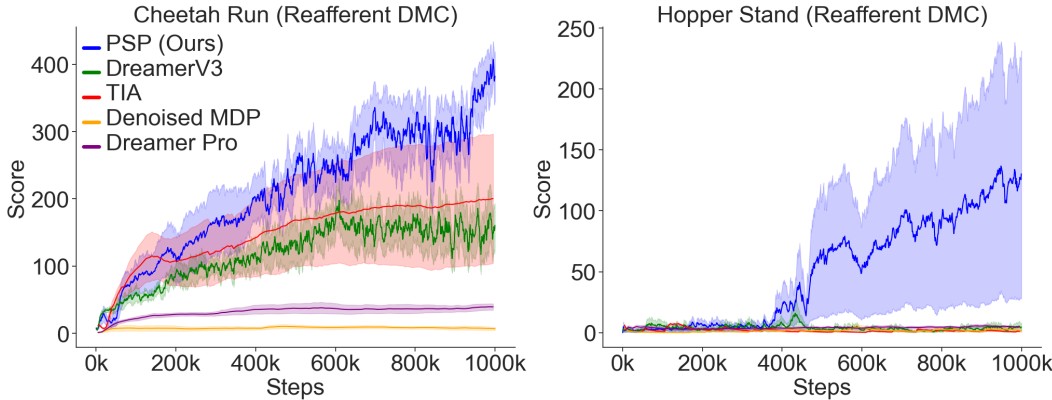

Figure 3: Training curve comparisons on Reafferent Deepmind Control. Mean ± std. err.

consisting of 2,500 16 x 16 grids, with each grid cell filled by a randomly chosen color. We then map a tuple of time (625 possible values) and a discretized version of the first action dimension (4 possible values) to an assigned background. We devised this to be analogous to the types of high complexity self-generated distractors found in the natural world (e.g. one's shadow).

This environment allows us to stress test the structural regularizations (and associated priors) that form the foundation of many existing distraction-avoidance methods. Many methods encode assumptions about the forms distractors will take (usually uncorrelated to agent actions, reward, or both), rather than a means of generally identifying and ignoring distractors. We hypothesize that a learning-based approach, in which we avoid distraction by learning what is actually important for the agent to get things done, has the potential to overcome even learnable-but-not-useful distractions.

We find that *none* of the baseline MBRL methods perform well on the Reafferent Environment, relative to their published results on the Distracting Control Suite or their performance on the unmodified Deepmind Control Suite (Table 1, Figure 3). We find that the world models learn excellent reproductions of the distracting background. However, the cost of this is that the reconstruction of the agent becomes less well-defined or even replaced by the background, especially in positions where the outcome of a movement is uncertain (see Figure 4 and A1 for examples with DreamerV3 and A2 with Denoised MDPs). This matches our expectation that the model will waste capacity on trivially predictable dynamics, rather than on the much more important but uncertain agent dynamics. As expected, unlearnable distractions are less challenging (Figure A3).

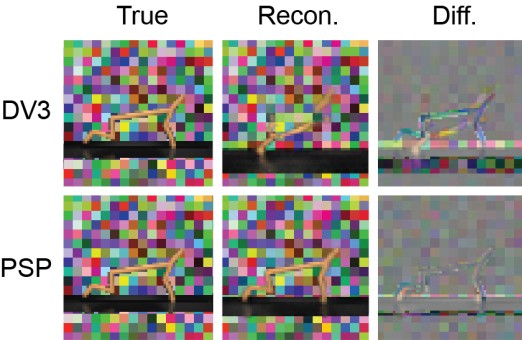

Figure 4: Reconstructed image comparison, PSP vs. DreamerV3 on Reafferent Cheetah Run, same episode and time point. True, reconstructed, difference (true - recon.). DreamerV3 accurately reproduces the background but not the cheetah.

Notably, the model-free DrQv2 agent shows a reduction in performance from its previous performance on the unmodified environment, but overall demonstrates quite robust performance (Table 1). This also matches our expectations, since the CNN encoder is learned as part of the policy in model-free learning, unlike with model-based, where the learning objective for the world model is separate from the learning objective for the policy.

Our new method demonstrates a substantial improvement over the existing baselines (Table 1, Figure 3). Although it shows a higher than desired level of variance between runs, especially on the more challenging Hopper Stand task, it nevertheless achieves scores beyond the reach of any of the baselines. We believe this **affirmatively answers Q1** by showing our new

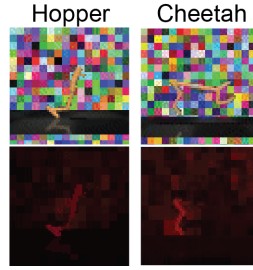

Figure 5: Example salience maps (policy-shaped loss weights) highlight the agent.

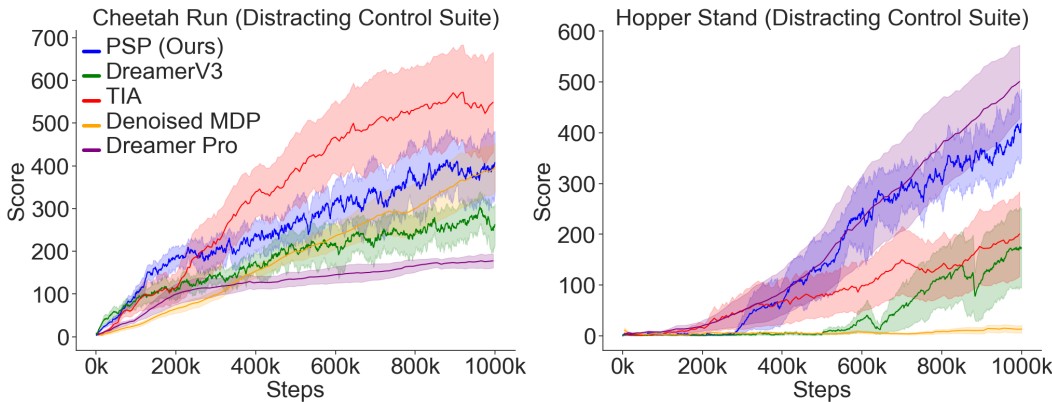

Figure 6: Training curve comparison on Distracting Control. Mean $\pm$ std. err.

agent is in fact robust to challenging distractors. **Addressing Q2**, we also find that the salience maps, derived from the gradient of the policy and used to weight the world model loss, highlight the regions of the image that we would expect (Figures 5 and A4). Interestingly, we see that sometimes the cheetah's rear leg is highlighted when it is the only leg close to the ground, though in other instances the entire cheetah is highlighted (Figure A4).

### 3.3 Performance on unaltered DMC and Distracting Control Suite

On Distracting Control tasks, in which the background distractor is uncoupled from the agent's actions, PSP produced consistently improved performance relative to baseline DreamerV3, in contrast to the more variable performance of DreamerPro, TIA, and Denoised MDP (Table 1, Figure 6). **This addresses Q3.**

Importantly, PSP also shows comparable performance to other methods (including DreamerV3) on the unaltered Deepmind Control Suite, demonstrating that we have not introduced a tradeoff between performance on distracting and non-distracting environments (Table 1, Figure A5), **resolving Q4.**

In sum, PSP exhibits similar performance to baseline methods in commonly used tests of distractor-suppression and in non-distracting environments, while also demonstrating unmatched performance on particularly challenging distractors that are complex but learnable.

### 3.4 Ablation study

To understand the contributions of each sub-component of the method (i.e. **address Q5**), we conduct ablations on the reafferent and unaltered Cheetah Run (Table 2). We find that some ablations trade off performance between the environments, while our complete model has good performance on both.

For instance, the top-performing method on the reafferent environment does not incorporate the segmentation or adversarial components, and uses the value gradient rather than the policy gradient. However, the variance of its scores is higher than any other approach, and more problematically, it shows the worst performance of all experiments in the unaltered environment. We believe this occurs

Table 1: Performance comparison across environments. DrQv2 is model-free, all others are model-based. TIA is task-informed abstraction, dMDP is denoised MDP, mean $\pm$ standard deviation.

| Task | DrQv2 | DreamerV3 | DreamerPro | TIA | dMDP | PSP |
|------|-------|-----------|------------|-----|------|-----|
| | | | Reafferent Control | | | |
| Cheetah Run | $565.1 \pm 35.5$ | $158.4 \pm 45.7$ | $39.7 \pm 9.0$ | $200.4 \pm 203.9$ | $6.7 \pm 4.3$ | $\mathbf{383.1 \pm 23.8}$ |
| Hopper Stand | $210.3 \pm 353.8$ | $4.6 \pm 3.9$ | $3.8 \pm 1.0$ | $0.9 \pm 0.3$ | $1.7 \pm 2.5$ | $\mathbf{128.5 \pm 215.7}$ |
| | | | Unmodified Deepmind Control | | | |
| Cheetah Run | $736.0 \pm 17.0$ | $521.1 \pm 136.3$ | $908.4 \pm 1.6$ | $773.7 \pm 22.7$ | $763.0 \pm 62.8$ | $712.3 \pm 32.3$ |
| Hopper Stand | $752.9 \pm 206.8$ | $867.4 \pm 15.9$ | $890 \pm 11.2$ | $298.4 \pm 512$ | $897.9 \pm 14.2$ | $865.6 \pm 53.6$ |
| | | | Distracting Deepmind Control | | | |
| Cheetah Run | $364.4 \pm 60.7$ | $243.8 \pm 81.2$ | $179.1 \pm 24$ | $548.5 \pm 238.9$ | $397.4 \pm 111.8$ | $408.6 \pm 125.1$ |
| Hopper Stand | $781.1 \pm 110.3$ | $173.7 \pm 160.9$ | $561.8 \pm 103.1$ | $200.5 \pm 171.7$ | $13.2 \pm 16.5$ | $417.7 \pm 118.9$ |

Table 2: Performance of ablated versions of PSP. Scores are shown for Cheetah Run in unaltered and reafferent Deepmind Control environments. The unablated PSP achieves good performance on both environments, and while some ablations achieve slightly better scores on either unaltered or reafferent, they trade off performance in the other environment.

| Gradient weighting | Gradient weighting with segmentation | Adversarial Action Head | Unaltered | Reafferent |
|---|---|---|---|---|
| Policy | ✓ | ✓ | $712.3 \pm 32.3$ | $383.1 \pm 23.8$ |
| Policy | ✓ | ✗ | $653.9 \pm 44.2$ | $231.2 \pm 58.6$ |
| Policy | ✗ | ✓ | $742.1 \pm 79.7$ | $188.4 \pm 9.4$ |
| None | ✗ | ✓ | $674.2 \pm 50.7$ | $324.4 \pm 2.3$ |
| Policy | ✗ | ✗ | $418.2 \pm 53.1$ | $379.0 \pm 46.8$ |
| Value | ✗ | ✗ | $381.7 \pm 64.9$ | $445.7 \pm 126.9$ |
| None | ✗ | ✗ | $521.1 \pm 136.3$ | $158.4 \pm 45.7$ |

because of the flaws in using only the gradient as an explanation for pixels that explain the actor-critic output. These flaws are more evident in an environment where the background never changes, as the policy is not required to learn any robustness to shifts in the background. In other words, the gradient for changes in the static background may be quite large, since the model is immediately out of domain when the background changes. We find that segmentation-based aggregation is critical to improving our model's performance amid distractors, while also maintaining its performance in the non-adversarial baseline. Overall, the results of the ablations confirm that combining segmentation, policy gradient sensory weighting, and adversarial action prediction results in the best scores across the unaltered and reafferent environments.

We also investigated the importance of the weight interpolation (Equation 3). We find that interpolation produces the expected benefit of allowing the agent to construct a useful world model, even when the policy is not very good, and thus sidestepping the 'chicken-and-egg' problem where the agent has neither a good policy nor a good world model (Figure A6). Furthermore, we tested the ability of PSP to adapt to either a task change (Figure A7) or a change in the distractor (Figure A8), and we found that PSP was able to quickly adapt in both scenarios.

## 3.5   Additional segmentation models

We additionally tested the sensitivity of PSP to the segmentation model. Given that segmentation models are likely to continue improving over time, we wondered 1) whether PSP could be compatible with other models besides SAM, and 2) how PSP performance might be modulated by the performance of the segmentation model. To investigate, we used the recently released SAM2 [Ravi et al., 2024], which has multiple model sizes that allow for trading off performance for segmentation speed, with as high as 6x faster segmentation speeds than the original SAM. We updated PSP to use the 'tiny' SAM2 model, the smallest and lowest accuracy of the provided model sizes. Our basic implementation with SAM2-tiny immediately improved segmentation

Table 3: Performance comparison of different segmentation models. SAM2 is the tiny model size of the new, faster Segment Anything model. SAM2-tiny performs similarly on Cheetah Run, but is worse on the more challenging Hopper Stand. SAM2-large recovers some of the decreased performance on distracting variants of Hopper. The full training plots are included in A10 and A11.

| Task | SAM | SAM2 (tiny) | SAM2 (large) |
|---|---|---|---|
| Reafferent Control | | | |
| Cheetah Run | $383.1 \pm 23.8$ | $398.7 \pm 70.5$ | - |
| Hopper Stand | $130.3 \pm 214.1$ | $29.4 \pm 45.3$ | $41.4 \pm 74.4$ |
| Unmodified Deepmind Control | | | |
| Cheetah Run | $712.3 \pm 32.3$ | $696.3 \pm 20.0$ | - |
| Hopper Stand | $865.6 \pm 53.6$ | $891.2 \pm 39.6$ | - |
| Distracting Deepmind Control | | | |
| Cheetah Run | $364.4 \pm 60.7$ | $352.0 \pm 60.4$ | - |
| Hopper Stand | $417.7 \pm 118.9$ | $187.4 \pm 172.0$ | $465.0 \pm 166.6$ |

speeds (and thus reduced the resources necessary for segmentation) by 2x. We found that PSP with SAM2-tiny yielded nearly identical performance as the original PSP with SAM on all three Cheetah environments (Table 3). On Hopper, a substantially harder task, we observed increased sensitivity to the quality of the segmentation. PSP with SAM2-tiny performed the same as PSP with SAM in the unmodified environment. In the Reafferent environment, PSP with SAM2-tiny still outperformed all baselines, with one of three runs yielding a successful policy (compared with zero out of twelve total runs across all baselines), though the successful run yielded a lower score (81.7)

than the successful run with SAM (377.5), which was also one of three runs. Additionally, SAM2-tiny yielded significantly worse performance on Hopper in Distracting Control. We hypothesized that this environment is particularly challenging for segmentation because the distracting background is a black and white video, which is difficult to discern from the platform. SAM2-tiny is less successful at segmenting the platform, which is problematic for learning a good policy. We further hypothesized that this issue would be resolved by improving the performance of the segmentation model. We tested this by using the SAM2-large model, and found that indeed this recovered performance back to the level of the original SAM, yielding a score of $465 \pm 166.6$. We additionally tested SAM2-large on Reafferent Control Hopper and observed a similar pattern with, again, one out of three runs yielding a successful policy with a score (114.4) that improved on SAM2-tiny. Optimizations, including using SAM2's video segmentation capabilities and better utilization of the GPU, would likely further improve segmentation speed. These results also suggest that as long as the segmentation is 'good enough' to properly segment the environment, PSP is not very sensitive to the segmentation algorithm. We also note that objects can be over-segmented into multiple segments without causing problems (Figure A9), and thus adequate segmentation is not a particularly stringent requirement.

## 4 Related Work

**Distraction-sensitivity of model-based RL**    Recent advances in Model Based RL (MBRL) including World Models [Ha and Schmidhuber, 2018], SimPLe [Kaiser et al., 2019], MuZero [Schrittwieser et al., 2020], EfficientZero [Ye et al., 2021], DreamerV1 [Hafner et al., 2019], DreamerV2 [Hafner et al., 2020], and most recently DreamerV3 [Hafner et al., 2023] have surpassed model-free RL in settings such as Atari, Minecraft, and Deepmind Control Suite. One deficiency of current MBRL algorithms is a susceptibility of the world model to become overwhelmed by easily predictable distractors, in part due to mismatch between the objectives of the policy (maximizing reward) and the world model (accurately predicting future states) [Lambert et al., 2020].

One line of work attempts to address the distractability of MBRL through structural regularizations. Deng et al. [2022] uses contrastive learning of prototypes instead of image reconstruction. Lamb et al. [2022] introduces the Agent Control-Endogenous State Discovery algorithm, which discards information not relevant to elements of the environment within the agent's control. Task Informed Abstractions (TIA) identifies task-relevant and task-irrelevant features via an adversarial loss on reward-relevant information [Fu et al., 2021]. Denoised MDPs extends TIA's factorization to include notions of controllability [Wang et al., 2022]. Clavera et al. [2018] use meta-learning and an ensemble of dynamics models. These works form a strong body of solutions, given prior knowledge of likely distractors, but they can struggle if a distractor does not fall into the designed regularizations.

A different approach instead learns what is important by using the actor-critic functions to scale the importance of various learned dynamics. VaGraM uses value gradients to reweight state reconstruction loss [Voelcker et al., 2022], building on Lambert et al. [2020] and IterVAML [Farahmand, 2018], but VaGram does not operate on visual tasks. Eysenbach et al. [2022] propose a single objective for jointly training the model and policy. Goal-Aware Prediction learns a joint representation of the dynamics and a goal, by predicting a goal-state residual, although they describe this approach as likely still susceptible to distractions [Nair et al., 2020]. Seo et al. [2023] decouples visual representations and dynamics via an autoencoder, improving the performance of Dreamer on tasks involving small objects. Value-equivalent agents [Grimm et al., 2020], such as MuZero [Schrittwieser et al., 2020] or Value Prediction Networks [Oh et al., 2017], construct a world model that only aims to represent dynamics relevant to predicting the value function, in contrast to methods such as Dreamer that aim to learn the broader dynamics of the environment. MuZero is very effective in settings with discrete actions such as Atari, Go, and chess. Adaptation to domains with complex action spaces such as Deepmind Control Suite [Hubert et al., 2021] have shown some success, however Dreamer-based agents that include image reconstruction for world model learning can still exhibit superior performance, and the image-related signals have been shown to be essential to their performance [Hafner et al., 2020]. Building on these methods, our work investigates how to combine the benefits of both image reconstruction and task-aware modeling, through policy-shaped image-based world modeling, by applying concepts from VaGraM to the image-based MBRL setting.

**Distraction-sensitivity of model-free RL**    A parallel track of Model Free RL (MFRL) has its own body of literature, with a leading method DrQv2 [Yarats et al., 2021a] used in this paper

for comparison. DrQv2 is an off-policy actor-critic RL algorithm that operates directly on image observations, using DDPG as the base RL algorithm alongside random shift image augmentation for image generalization and sample efficiency. Although our work focuses on a solution to MBRL's distraction-sensitivity, it is worth noting analogous deficiencies can exist in MFRL and there are a number of works addressing these. For instance, Mott et al. [2019] uses an attention mechanism to make the agent robust to environment changes, and Tomar et al. [2024] learn task-relevant inputs mask. Yarats et al. [2021b], an inspiration for DreamerPro, creates prototypes that compress the sensory data, Grooten et al. [2023a] apply dynamic sparse training to the input layer of the actor and critic networks, and Grooten et al. [2023b] mask out non-salient pixels based on critic gradients.

## 5   Discussion

PSP combines three ideas to focus the capacity of an agent's world model on aspects of the environment that are useful to its policy. First, the gradient of the policy with respect to the input image is used to identify pixels that influence the policy. Second, the importance of individual pixels are aggregated by object, using a segmentation model to identify objects. Third, wasteful encoding of the preceding action (which is known and does not need to be predicted) in the image embedding is removed using an adversarial prediction head. Together, these allow an agent to construct a world model that best informs its policy, and in doing so, use the policy to shape what information is prioritized by its world model. The outcome of this process is an agent that is selective about what parts of the world it models, and that becomes resilient against enticingly learnable, but ultimately empty, distractions.

Our work draws a connection between the use of the value function gradient in VaGraM and related concepts from the vision model explainability literature. The value gradient can be seen as analogous to saliency maps [Simonyan et al., 2013]. Other gradient-based attribution methods, such as those that multiply saliency maps by input intensities [Shrikumar et al., 2017] or Integrated Gradients [Sundararajan et al., 2017, Ancona et al., 2019] offer additional ways to perform attribution. Some gradient-based attribution methods, such as Integrated Gradients, can be computationally expensive due to the need to approximate an integral over the input space. Future work may investigate incorporation of more advanced explainability methods such as these into PSP, and the concept of an agent 'interpreting itself' may exhibit broader utility. Finally, recent work that uses SAM combined with human supervision to improve the generalizability of model-free RL [Wang et al., 2023], together with our work, point towards the potential value of incorporating powerful object segmentation models into reinforcement learning systems.

**Limitations**   Limitations of PSP include its fundamentally object-centric view, which assumes that pixels belong to single objects, and that the objects can be ranked by their importance. Additionally, the SAM segmentation model requires significant compute, but these models will likely improve over time and can also be application-tailored. Notably, however, segmentation is not necessary during inference of the world model and policy, only training. Finally, it is not yet clear how well PSP will adapt in environments where the reward structure or salient features change across time. PSP may make the world model more task-specific than other approaches, although it does keep some reconstruction weight on non-task-relevant features and we observed initial evidence of resiliency (Figures A7, A8).

**Outlook**   Our work finds headroom to improve the robustness of MBRL to distractions by linking the actor-critic functions and the reconstruction loss and leveraging useful priors from pre-trained foundation models. The findings here open other lines of inquiry such as using better model explanation techniques or more explainable architectures, utilizing faster segmentation models, and utilizing segmentation models designed for videos, in order to do temporal aggregation. Substantial work likely remains to improve the speed of this technique and find extensions that allow it to reliably work for harder problems, such as applied robotics. The adversarial action prediction head's inspiration from the biological concept of efference copies also suggests there is still space in MBRL to consider biological metaphors as helpful design principles for learning algorithms. In sum, we present PSP, a method for avoiding distractors by focusing the world model on the parts of the environment that are important for selecting actions.

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

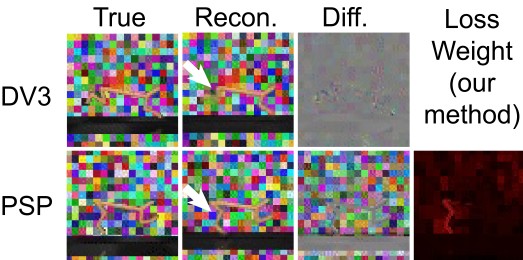

Figure A1: Reconstructed image comparison, PSP vs. DreamerV3 on Reafferent Cheetah Run. This highlights how PSP focuses on modeling the hind leg (white arrow), while DreamerV3 focuses on modeling the background but fails to model the hind leg. From left: True, reconstructed, difference (true - recon.), PSP salience loss weight.

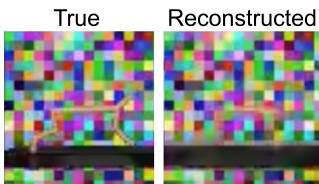

Figure A2: Denoised MDP reconstructs the background with a high degree of fidelity, but does not clearly render the Cheetah agent.

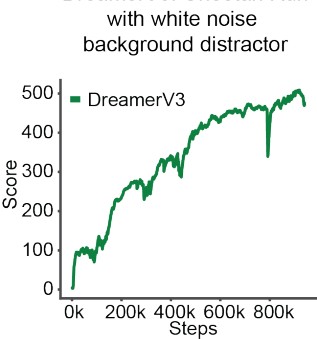

Figure A3: To more thoroughly show that the reafferent environment impacts DreamerV3 because of the learnable time & action mapping to backgrounds (and not purely because of the presence of the backgrounds themselves), we include this training curve from an environment which uses the same backgrounds, but with a random choice of background at each timestep. This demonstrates effective policy learning in spite of the distracting (but unlearnable) background.

## A  Broader Impacts

At the current stage, this work remains reasonably far from any large societal impacts, as it is limited to agents interacting with small, simulated environments. Over the long term, however, if model-based RL algorithms are used to control robots or internet-connected agents (such as large language model agents), the potential for both large positive and negative societal impacts becomes relevant. On the positive side, intelligent agents that are capable of modeling the world and avoiding distractors have the potential to aid humans in a wide variety of scenarios, from housework, to medical applications, to exploration, to internet research. On the negative side, agents without proper safeguards have the potential to inflict harm on humans and the environment, whether through negligence or malfeasance. Ultimately, our work is targeted at producing the positive impacts, while still allowing for mitigation of the negative impacts.

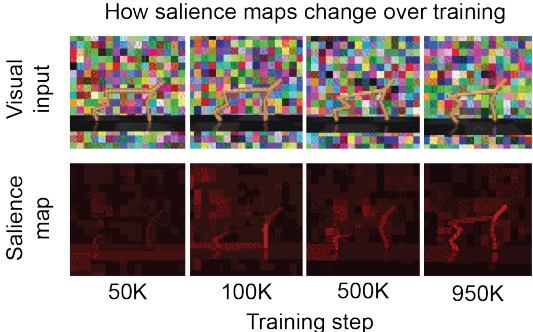

Figure A4: Improvement in specificity of the salience map across training.

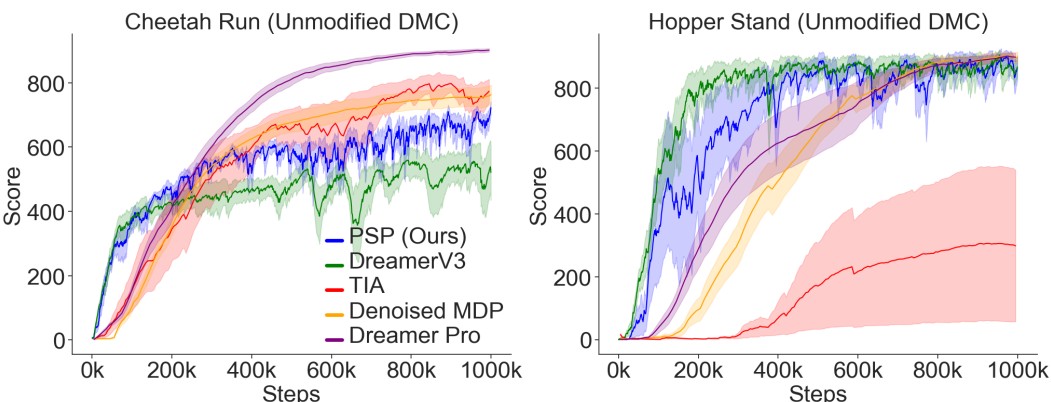

Figure A5: Training curve comparison on unmodified Deepmind Control. Mean ± std. err., n=3 runs.

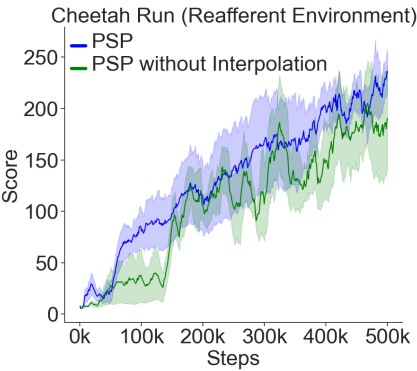

Figure A6: Interpolation of the gradient based reconstruction loss and a uniform reconstruction loss yields superior performance to a gradient based reconstruction loss without interpolation, which manifests as a faster initial rise in score with PSP. Interpolation allows the agent to construct a useful world model even when the policy is not yet very good (and hence minimizes the impact of the 'chicken-and-egg' problem where the agent has neither a good policy nor a good world model). Mean ± std. err., n=3 runs.

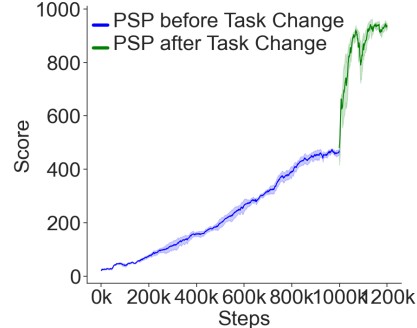

Figure A7: In order to test the adaptability of a model trained for one task via PSP to another task for the same agent, we switched from Walker Run to Walker Stand at step 1M, while leaving the Reafferent background unchanged. We see that the learned model can quickly adapt to the new task, even with the Reafferent background. Mean ± std. err., n=3 runs.

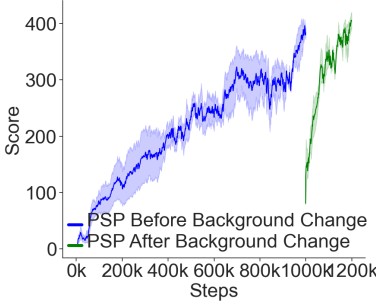

Figure A8: In order to test the adaptability of a PSP model to a dynamic distraction, we switched the settings of the Reafferent distraction at step 1M. We see that the learned model can quickly adapt to the new distraction. Mean ± std. err., n=2 runs.

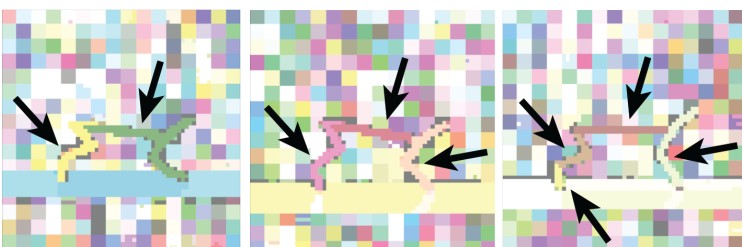

Figure A9: How high quality must the segmentation be? We note while training that SAM did not have perfect segmentation performance on our tasks. In fact, the Cheetah agent was usually segmented into several (and a varying number of) different regions. The algorithm proves robust across these segmentation variations.

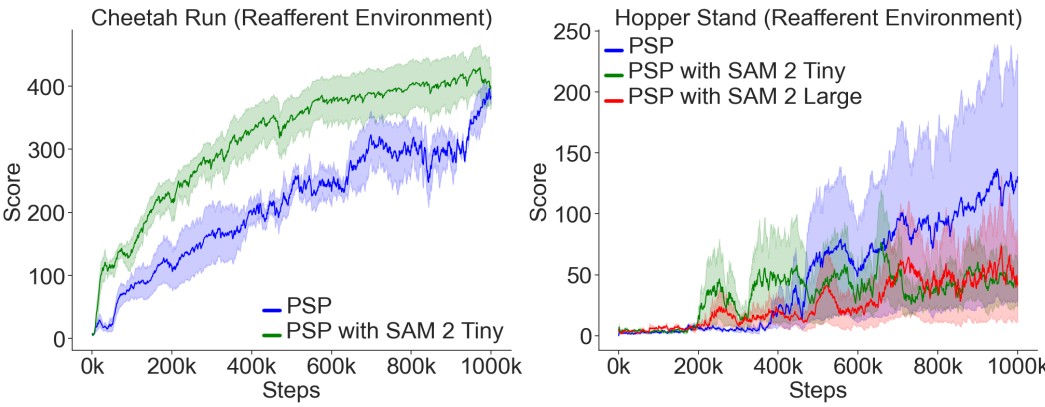

Figure A10: Training curve comparisons on Reafferent Deepmind Control. Mean ± std. err. of PSP with different segmentation algorithms: SAM, SAM 2 Tiny, and SAM 2 Large. All baselines are shown in Figure 3. Notably, all three PSP agents shown here achieve non-zero scores on Hopper Stand, in contrast to the baselines.

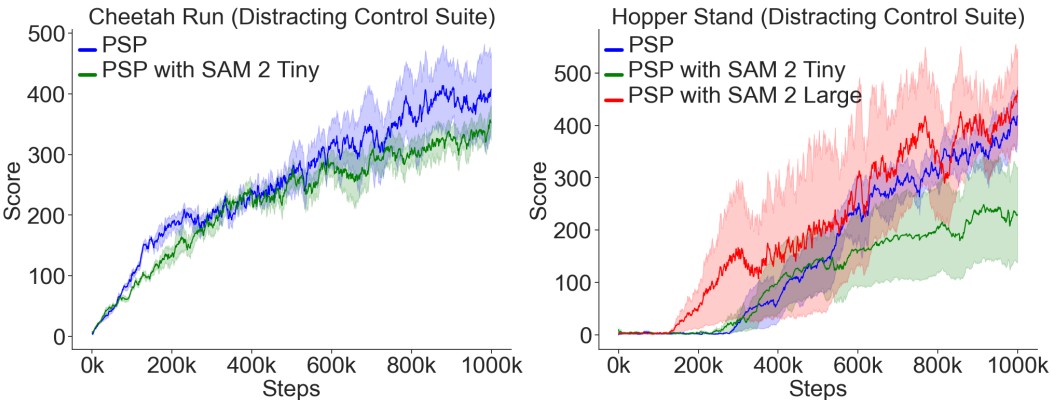

Figure A11: Training curve comparisons on Distracting Control Suite. Mean ± std. err. of PSP with different segmentation algorithms: SAM, SAM 2 Tiny, and SAM 2 Large. All baselines are shown in Figure 6.

## B  Computational Overhead

We characterize the computational cost of the PSP algorithm by ablating various components and measuring its training speed (Table A1). There are four major components of PSP that affect performance. In order of decreasing computational cost: (1) Policy gradient-based weighting, (2) image segmentation (e.g. with SAM) (3) action adversarial head, and (4) segmentation-based aggregation of the gradient weighting. These are all costs that apply only during training, not during inference.

**Policy gradient based weighting:** For each latent state produced by the encoder RSSM in each step of a rollout, we take the gradient of the policy with regard to the image pixel inputs. In our implementation, this auto differentiation yields a complexity of $\mathcal{O}((E + R + P) \cdot S^2 \cdot W \cdot H)$, where $E$ is the number of encoder parameters, $R$ is the number of parameters of the RSSM, $P$ is the number of parameters of the policy, $S$ is the number of steps in a rollout, $W$ is the width of the input image, and $H$ is the image height.

There are a couple of major opportunities for optimization in future implementations. First, for debugging and experimental reasons, we have been computing the gradient of the policy with respect to all rollout steps, instead of just the current step. This reveals an opportunity to reduce the computational burden by a factor of $S$, where $S$ is 64 in our implementation. Reducing this contribution could induce a significant speedup in the performance of future implementations. Second, we take the gradient of the policy with regard to the image during rollouts only to enable visualization

Table A1: Comparison of computational overhead. FPS stands for frames per second.

| Adv Head? | SAM? | Policy VAML? | Interpolation? | FPS |
|---|---|---|---|---|
| Yes | Yes | Yes | Yes | 5.0 |
| Yes | No | Yes | Yes | 5.9 |
| Yes | No | No | N/A | 17.1 |
| No | No | No | N/A | 19.0 |

for debugging and figures. This is strictly speaking unnecessary overhead and could be eliminated for a gain during training.

**Image segmentation:** The details of the additional complexity depend on the specific algorithm used. With our addition of results using the SAM2 tiny model, we now demonstrate the viability of using different segmentation algorithms. Notably, the segmentation process is entirely parallelizable separately from the training process: images are segmented as they are collected and stored in the replay buffer. We find that segmentation is not a bottleneck in training speed. Moreover we find that advances in segmentation algorithms (e.g. SAM2) reduce the computational resource requirements for this process.

**Adversarial head:** For each training step, we take the gradient of the loss of the action prediction head with regard to the parameters of the encoder. Therefore, the cost is $\mathcal{O}((E + R + A) \cdot S)$, where $A$ is the number of parameters of the action prediction head. These gradients are added with the regular gradients during a train step, so the adversarial head does not introduce additional iterations during training.

**Segmentation aggregation of gradients:** We take the mean value of the gradient with respect to each mask by multiplying each mask by the gradient weighting and then taking the hadamard product of the resulting image (matrix) with the inverse of the sum of mask elements equal to 1. This results in an additional cost of $\mathcal{O}(M \cdot W \cdot H)$, where $M$ is the number of masks.

## C   Experiments Compute Resources

Each trial of the PSP method used 4 Nvidia A40 GPUs to train the modified DreamerV3 model, and 4 A40 GPUs to run the Segment Anything model in parallel. Given an estimated 17 unique experiments for the final paper, 3 trials per experiment with our method, and about 1.5 days per training run, we used about 17 * 3 * 1.5 * 8 GPUs = 612 GPU days on A40 accelerators. Early experiments with this methodology likely used an additional 300. Baseline trials could be run on only a single A40 GPU or a desktop NVIDIA 2070 SUPER, usually in less than a day, and accounted for a comparably negligible level of resources.

We believe this level of resource consumption could be easily reduced. The modifications to the DreamerV3 model do not attempt to benchmark the most costly components. We suspect our method of parallelizing the new backpropagation from the policy to the image could be optimized further from its naive Jax implementation. Additionally, SAM could be supplanted by a more efficient segmentation model. We focused on establishing the basic technique with SAM, and replacing it with more efficient methods should be the subject of future work.

## D   Code

The repository with code and instructions for reproducing these experiments is available at this GitHub Repository.

