# OpenReview forum: "Policy-shaped prediction: avoiding distractions in model-based reinforcement learning"
_NeurIPS.cc/2024/Conference — NeurIPS 2024 poster_

### Official Review · Reviewer_aBfS · 2024-07-04

**Soundness:** 3
**Presentation:** 3
**Contribution:** 3
**Rating:** 6
**Confidence:** 4

**Summary:**

The paper addresses the problem of distractions in model-based RL by proposing a policy-shaped prediction (PSP) method, which combines segmentation and adversarial learning to accurately identify and prioritize policy learning on crucial parts of dynamics by incorporating saliency maps from image-based environments. Authors achieve this by finding relevant importance signals from the gradients of the policy w.r.t input images and aggregate importance signal into gradient-based weighting by SAM network. As the main motivation, along with the new distraction-suppression algorithm, the paper introduces a novel distraction benchmark. Overall, the authors show that the proposed PSP demonstrates up to 2x improvement in robustness from distractions, while attaining similar scores in distraction free settings as baselines.

**Strengths:**

The motivation behind proposed approach is sound and clearly explained.

Empirical results support the main claims of the paper, proving the efficiency of proposed method.

Main novelty stems from enhancing the application of policy-gradient based weighting of the important features by reformulating weight factor in terms of mean of different segmentation masks, provided from powerful segmentation models.

Detailed description of novel distraction benchmark is provided, showcasing the ability of PSP to be robust against static adversarial perturbations, surpassing previous methods.

Authors provide concise and dense experiments, ablation studies, along with all the necessary reproducibility details.

**Weaknesses:**

1)	It is not clear whether much simpler segmentation models would work, because the method seems to be agnostic to the choice of the segmentation model. Evaluation of additional segmentation models would support the paper and could potentially increase the overall speed of PSP training.
2)	In the new benchmark description, Section 3.2, the authors claim that the background distractions are predictable and dependent on the agent’s actions and dynamics. However, the most natural distractions are independent from the agent and ignoring such dynamic case greatly limits the usefulness of the proposed benchmark.
3)	The paper does not mention domain adaptation approaches in RL, which solve exactly same task of learning general policy, invariant to distractions or other kinds of domain shifts.

**Questions:**

1)	It would be beneficial to include images of saliency maps during training and to discuss how they vary.
2)	Could the authors change SAM in PSP to another segmentation model? How would the final results change?
3)	How well the algorithm performs on dynamically changing distractions? (e.g noisy TV)
4)	How accurate are the SAM masks? How said inaccuracies affect the weight factor in Eq (2) on the overall objective in Eq (1)?
5)	Could the proposed method be applied in model-free RL setting?
6)	What are the benefits of using PSP against domain adaptation algorithms, which are also robust to perturbations as they learn general invariant representations of agent dynamics/environment (known to be robust to domain shifts)? (e.g., refer to Raychaudhuri et al, "Cross-Domain Imitation Learning from Observations")
7)	In Table 2, the results for Reafferent environment with gradient weighting and with the adversarial action head seem to be very similar to the case when both of those are removed (383.1 +- 23.8 and 379.0 +- 46.8). Could the authors provide some reasoning for why is this the case?

**Limitations:**

Authors fully addressed possible limitations and potential societal impacts in Section 5 of the paper.

---

> ### Author Rebuttal · Authors · 2024-08-07
>
> We thank the reviewer for their excellent comments.
>
> **Evaluation of additional segmentation models**:
> Thank you for this great suggestion. We now include evaluation the 'tiny' variant of the recently released SAM2 segmentation model. This increases the speed of the segmentation step by over 2x (in a very unoptimized implementation), and yields similarly performant results (Fig. R1A).
>
> **Dependent vs. independent distractions**:
> Existing benchmarks such as Distracting Control Suite target the described scenario -- with distractions that are independent from the agent, and natural videos as the distracting background. The goal of our Reafferent environment is to complement these existing benchmarks by focusing on scenarios where distractions are predictable and dependent on the agent. The Reafferent environment is not meant to replace the Distracting Control Suite, but rather to augment it in an important and challenging way.
>
> **Relation to domain-adaptation algorithms**:
> Thank you for bringing up the interesting connection to domain adaptation algorithms. In general, the domain adaptation setting is somewhat different from our problem: in domain adaptation, (e.g. Raychaudhuri et al) there is an expert environment (which in our case might be an environment without distractions) and an agent environment (perhaps with distractions). The task is to adapt policies from the non-distracting environment to the distracting environment. This is an interesting and potentially fruitful approach, but it is also quite different from our setting in which the distractor is potentially always present. Additionally, even if the setting were directly applicable, Raychaudhuri et al. only uses a low dimensional joint-level state, and it is unclear how or if that would scale to high dimensional visual observations. Nevertheless, we will add discussion of the conceptual relevance of domain adaptation to the manuscript. Thank you.
>
> **Images of saliency maps during training**:
> Thank you for the great suggestion, we now show example saliency maps from across training (Fig. R1F). Qualitatively, the saliency maps appear to improve their specificity across training.
>
> **Dynamically changing distractions (e.g. noisy TV)**:
> Thank you for raising the interesting scenario of a noisy TV. Interestingly, existing algorithms such as baseline DreamerV3 actually have no problem dealing with such distractions: because the distraction is not predictable, the model learns to simply ignore it. Thus as the distractions become more dynamically changing and less predictable, they may tend to become less problematic. In Fig. R1K, we provide an example training curve from a baseline DreamerV3 agent in a cheetah environment with randomly-changing background distractions, showing that it is able to succeed in the face of this white noise distraction.
>
> **Accuracy of SAM masks**:
> We investigated this important question in two ways. First, we implemented PSP with the tiny variant of SAM2, and observe similar results as with SAM. Second, we investigated the segmentation masks directly and observe evidence that precise details of the segmentation may not matter much.  In Fig. R1B for example, in a performant PSP agent, the cheetah is sometimes split in two, three, or four segments, which suggests a degree of insensitivity to exact details of the segmentation. Thus, PSP appears to be somewhat insensitive to the exact details of the segmentation. Indeed, because the segmentation only acts as a way to average together gradient signals (Eq. 2), so as long as it groups things of relatively similar importance, the precise details and exact accuracy do not appear to matter much.
>
> **Application to model-free RL**:
> This is an interesting question, thank you. In model-free methods the latent state is already learned based on the gradient signals from the policy and value function, in contrast to the model-based settings (such as Dreamer) where the latent state is also informed by a reconstruction loss. Indeed, as we show, the model-free baseline DrQv2 performs quite well in the distracting settings. Our work is focused on bringing some of the advantageous features of the model-free approach into the model-based realm.
> However, it is quite possible that incorporating segmentation via an additional channel in the input image or adding an action-prediction head could clean up the image embedding in the model-free RL setting. This would be very interesting to investigate in future work.
>
> **Ablation results**:
> This is a great observation that some of the ablation results appear to similar to each other. This observation touches on three key points.
> First, the latter score (379.0) *does* have policy gradient weighting, it just does not use the segmentation model to aggregate gradients. The comparison with no gradient weighting and no action head would be the unaltered DreamerV3 of Table 1 (which scores substantially worse, 158.4 ± 45.7), which we have now included in the ablation table.
> Second, while there is a similar score in both cases on the Reafferent environment, there is a substantially better score with the full PSP approach on the Unaltered environment (712.3 vs 418.2).
> Third, while the gradient signal alone is adequate to boost performance on the reafferent environment, the gradient signal alone is noisy and this can disrupt the higher scores that are attainable on the unaltered environment. By incorporating the additional methods, especially the segmentation, we can reduce the noisiness of the gradient signal. We now include an additional ablation: policy gradient weighting + segmentation but without the action head (Fig. R1I), showing that the segmentation improves performance on the unaltered environment (compare the second and fifth row). Combining the gradient signal, segmentation, and adversarial action prediction as PSP then yields good performance on both environments.

---

> > ### Comment · Reviewer_aBfS · 2024-08-09
> >
> > I appreciate all the clarifications. As a reviewer who gave the highest score to this work, I do *not* think I should be increasing the score any higher. For the discussion period, however, I do want to underscore that the authors conducted a lot of new meaningful experiments for their rebuttal. On the other hand, the response to 1AxW on the added complexity appears to be too empirical, without any attempts of assessing the complexity analytically (even if taking into account a sound ablation effort). Hence, I keep the original score.

---

> > > ### Author Response · Authors · 2024-08-09
> > > **Response to aBfS response**
> > >
> > > ### Overview
> > > We deeply appreciate the reviewer for taking the time and effort to review our additional results, and we thank this reviewer for noting the meaningful addition these results contribute to the work. **We believe we have addressed each of this reviewer's stated concerns**, including with three new experiments (SAM2, noisy TV, and additional ablations) and additional data (example saliency maps, example segmentations). Furthermore, we believe these *new and expanded result address most of the concerns of all reviewers* and should be accounted for if considering others' scores when determining whether to raise this reviewer's score. Thank you again for your effort and for your thoughtful and constructive comments.
> > >
> > > ### Analytical complexity
> > > In terms of the response to 1AxW, we are happy to discuss the analytical properties of PSP. There are four major components of PSP that affect performance. In order of additional compute: (1) Policy gradient based weighting, (2) image segmentation (e.g. with SAM)  (3) action adversarial head, and (4) segmentation-based aggregation of the gradient weighting.
> > >
> > > *Policy gradient based weighting*: For each latent state produced by the encoder RSSM in each step of a rollout, we take the gradient of the policy with regard to the image pixel inputs. This auto differentiation yields a complexity of O((E + R + P)\*S^2\*W\*H), where E is the number of encoder parameters, R is the number of parameters of the RSSM, P is the number of parameters of the policy, S is the number of steps in a rollout, W is the width of the input image, and H is the image height.
> > >
> > > There are a couple of major opportunities for optimization in future implementations. First, for debugging and experimental reasons, we have been computing the gradient of the policy with respect to all rollout steps, instead of just the current step. This reveals an opportunity to reduce the computational burden by a factor of S, where S is 64 in our implementation. We note reducing this contribution could likely induce a radical speedup in the performance of future implementations. Second, we take the gradient of the policy with regard to the image during rollouts only to enable visualization for debugging & figures. This is strictly speaking unnecessary overhead and could be eliminated for a gain during training.
> > >
> > > *Image segmentation*: The details of the additional complexity depend on the specific algorithm used. With our addition of results using SAM2 tiny, we now demonstrate the viability of using different segmentation algorithms. Notably, the segmentation process is entirely parallelizable separately from the training process: images are segmented as they are collected and stored in the replay buffer. We find that segmentation is *not* a bottleneck in training speed. Moreover we find that advances in segmentation algorithms (e.g. SAM2) reduce the computational resource requirements for this process.
> > >
> > > *Adversarial head*: For each training step, we take the gradient of the loss of the action prediction head with regard to the parameters of the encoder. Therefore, the cost is O((E + R + A)*S), where A is the number of parameters of the action prediction head. These gradients are added with the regular gradients during a train step, so the adversarial head does not introduce additional iterations during training.
> > >
> > > *Segmentation aggregation of gradients*: We take the mean value of the gradient with respect to each mask by multiplying each mask by the gradient weighting & then taking the hadamard product of the resulting image (matrix) with the inverse of the sum of mask elements that are 1. This results in an additional cost of O(M\*W\*H), where M is the number of masks.
> > >
> > > While these analytical properties can guide further improvements to the algorithm, we ultimately note that the set of hardware and software used is a very important component of actual wall clock performance. Many elements of complexity are parallelizable on a GPU, and thus may not actually contribute greatly to wall clock time. Additionally, Jax & CUDNN make choices about the exact algorithms that implement the high level graph, which in our experience can impact not only wall clock time but even numerical stability. Therefore, we do believe that any assessment of performance must include concrete code and empirical assessments, and this is why we focused on presenting empirical comparisons. Thank you for the opportunity to clarify.

---

### Official Review · Reviewer_1AxW · 2024-07-05

**Soundness:** 3
**Presentation:** 3
**Contribution:** 3
**Rating:** 5
**Confidence:** 3

**Summary:**

This paper introduces Policy-Shaped Prediction (PSP), a method in model-based reinforcement learning (MBRL) designed to focus on significant aspects of an environment by reducing the influence of distracting information. PSP incorporates a pre-trained segmentation model, a task-aware reconstruction loss, and adversarial learning to enhance the robustness and focus of MBRL agents. The method is evaluated using a benchmark tailored to assess its efficacy against intricate and predictable distractions that do not aid in decision-making.

**Strengths:**

- The introduction of a segmentation model to MBRL is innovative and appears effective based on the results presented.
- The paper demonstrates improvements over existing approaches in terms of focusing learning on relevant environmental features and maintaining performance in distraction-filled settings.
- Methodology extends well-known concepts with new mechanisms, potentially setting a foundation for further exploration in distraction-resistant MBRL.

**Weaknesses:**

- PSP exhibits high variance in performance, particularly noted in tasks like Hopper Stand, indicating that the method may not consistently achieve the best results across all scenarios.
- The implementation of PSP, especially with the use of segmentation models like SAM and adversarial learning, is resource-intensive, which might limit its practical applicability in environments with limited computational resources.
- The added complexity of PSP, including policy-gradient weighting and adversarial components, might pose challenges for implementation and stability during training.

**Questions:**

- Given the resource-intensive nature of PSP, how feasible is it to apply this method in real-world applications, such as robotics or autonomous driving, where computational resources and real-time performance are critical?
- How does PSP perform in dynamic environments where the reward structure or salient features change over time? Is the method adaptable to such scenarios?
- How sensitive is PSP to the quality of the segmentation model used? Would a less accurate segmentation model significantly impact the performance of PSP?

**Limitations:**

While the PSP shows impressive performance in the introduced benchmark, its dependency on a high-quality segmentation model could limit its deployment in varied or less-controlled environments. Furthermore, the paper lacks a detailed discussion on the potential computational overhead introduced by PSP components.

---

> ### Author Rebuttal · Authors · 2024-08-07
>
> We thank the reviewer for their valuable comments.
>
> **High variance in Hopper Stand:** Thank you for this observation. Hopper Stand is a challenging environment with fairly sparse reward. Indeed none of the other MBRL methods were ever able to achieve success on this task in the Reafferent environment (the maximum score of any method across this grand total of 12 runs was 8.7), whereas PSP achieved a score as high as 377.5. Thus, while all baseline methods consistently exhibited low performance on this task, across all runs in our experiments, PSP was uniquely able to reach a higher score.
>
> Additionally, we note that we actually had a fourth run of PSP on Hopper Stand with Reafferent background that was quite promising but crashed (irrecoverably) before completing 1M steps, and was therefore not included in our tally, which achieved a score 389 by step ~900K. Thus, 2/4 runs with PSP yielded an agent that figured out how to successfully stand, whereas 0/12 MBRL baseline runs did (Fig. R1J).
>
> **Applicability in environments with limited computational resources:**
> Thank you for this great point. We highlight that the added burden of PSP only occurs during training, and not policy inference, which is unchanged. Additionally, the adversarial learning, on its own, does not introduce significant additional computational burden, as the action head is much smaller than the image encoder. We see that the ablation in which only this head is added takes only an additional 11% training time (Fig. R1L).
>
> Regarding the expense of applying SAM, we acknowledge the burden this model introduces. We initially chose SAM as it was most likely have good segmentation quality across environments. However, this model has also spawned a number of cheaper segmentation approaches that could make this approach far less resource-intensive. In particular, the recently released SAM2 can achieve >6x inference speeds relative to SAM (Table 6 of Ravi et al 2024) and thus can reduce the required resources. We expect metrics like this to further improve as models are distilled, optimized, and improved. As highlighted before, we have now included results with the 'tiny' variant of SAM2, which exhibits similar PSP performance to SAM (Fig. R1A).
>
> **Complexity adding challenges for implementation and training stability**:
> Thank you for raising these points. Empirically, we observe that PSP is stable on Cheetah Run. It is less stable on Hopper Stand, but this is a challenging environment in which the baselines are substantially less performant, and PSP does not appear to be less stable than the baselines.
>
> In terms of implementation complexity, we introduce several important changes, which our ablations show to be critical. Our implementation is therefore admittedly more complex than the baseline. However, the changes primarily consist of adding a few extra terms to the loss function and incorporating a segmentation module, and the reference implementation we provide demonstrates that it is fairly straightforward to make these additions to an algorithm such as DreamerV3.
>
> **Feasibility for application to real-world**:
> We appreciate the reviewer's interest in real-world applications, and we are excited about future work investigating such applications. We wish to highlight that the inference cost of a model trained with PSP is identical to its inference cost without PSP training; only the training cost is impacted. Therefore, any model-based RL method that is feasible for such applications already would still be feasible with the addition of PSP training techniques. We moreover point out that in general it is not trivial to apply Dreamer-type algorithms in real-world settings, but that there is encouraging precedent (e.g. DayDreamer, Wu, Escontrela, Hafner et al. 2022)
>
> **Dynamic environments with changing reward structure or salient features**:
> We expect that PSP is likely capable of adapting to a changing reward structure, especially if the salient features remain the same. To test this, we trained a PSP agent on Walker Run for 1M steps, and then switched to a Walker Stand reward. We found that PSP was able to quickly adapt to the new reward function (Fig. R1D). Changes in salient features represent a more difficult challenge, but we think that PSP would still be able to adapt, in part because of the flexibility afforded by the mask interpolation of Eq. 3 (see Fig. R1C). It is likely that PSP will adapt more slowly then an algorithm that does not selectively prioritize parts of the scene (though the latter algorithms will not be robust to distractions). Methods such as Curious Replay (Kauvar et al 2023) may help mitigate these adaptation challenges.
>
> **Sensitivity to quality of segmentation model**:
> We investigated this important question in two ways. First, we implemented PSP with the tiny variant of SAM2, and observe similar results as with SAM (Fig. R1A). Second, we investigated the segmentation masks directly, and we observe evidence that precise details of the segmentation may not matter much. In particular, as shown in Fig. R1B, the SAM segmentation does *not* segment the cheetah perfectly. In some cases, the cheetah is split into 2 segments, while in other cases it is split into 3 or 4 segments. Thus, PSP appears to be somewhat insensitive to the exact details of the segmentation.
> In terms of deployment to varied or less-controlled environments, we again highlight that the segmentation is only used during the training step (which could be offline), and is not used during inference/deployment of the trained model/policy.
>
> **Computational overhead**:
> Thank you for this suggestion. We have now detailed a break down of the computational overhead of the PSP components (Fig. R1L).

---

> > ### Comment · Reviewer_1AxW · 2024-08-12
> > **Official Comment by Reviewer 1AxW**
> >
> > Thank you for your detailed and thoughtful rebuttal. I have carefully reviewed your responses, including the additional experiments and clarifications provided in response to Reviewer aBfS's comments. I appreciate the analytical breakdown of PSP's complexity and the steps taken to mitigate computational overhead, especially regarding the segmentation process with SAM2. However, I concur with Reviewer aBfS that further analytical consideration of the empirical complexity is needed to better understand its impact on broader adoption, particularly in resource-limited environments. While your response and contributions are valuable, I will keep my original score, as addressing these complexity concerns more thoroughly would strengthen the paper’s contribution.

---

> > > ### Author Response · Authors · 2024-08-12
> > > **Clarification request**
> > >
> > > We sincerely appreciate the reviewer's time and engagement with our responses. Thank you. We welcome further discussion and would like to seek clarification on a specific point. Could the reviewer please elaborate on what is meant by an "analytical consideration of the empirical complexity"?
> > >
> > > Our experiments demonstrated significantly improved performance over the baseline DreamerV3 in distracting environments while maintaining performance in less-distracting settings. This considerable increase in performance only reduced the unoptimized wall clock speed by less than a factor of four. While optimization efforts could likely improve this speed, we believe the novel ideas and performance improvements introduced in our work are the primary focus.
> > >
> > > Furthermore, we showed that ongoing innovations in segmentation algorithms yield equivalent performance on PSP while significantly reducing computational resource requirements.
> > > We would greatly appreciate if the reviewer could specify what additional information they are seeking beyond what we provided, including in our response to aBfS. Your insights are very valuable to us, and we look forward to addressing any remaining concerns or questions.

---

> > > > ### Comment · Reviewer_1AxW · 2024-08-14
> > > >
> > > > Thank you for your detailed and thoughtful rebuttal. I want to start by acknowledging the significant contributions of this paper. The introduction of PSP is indeed an innovative approach to addressing distractions in model-based reinforcement learning, and the results you have demonstrated, particularly in challenging environments, are impressive. My current score reflects an appropriate evaluation of the paper's contributions and its potential impact on the field.
> > > >
> > > > That said, there are a few remaining concerns, particularly regarding what I meant by an "analytical consideration of the empirical complexity." To clarify, my request pertains to a deeper exploration of how the computational complexity of the proposed PSP approach scales across different environments and hardware setups. While the empirical results you provided give insight into the performance under specific conditions, I believe a more detailed analysis could include:
> > > >
> > > > 1. **Scalability Analysis**: How does the complexity of each component of PSP (e.g., segmentation, adversarial learning) scale as the problem size increases, particularly in terms of state space, action space, and environment complexity? This could involve theoretical analysis or case studies showing performance across various levels of task complexity or hardware capabilities.
> > > >
> > > > 2. **Trade-off Analysis**: An exploration of the trade-offs between segmentation accuracy, computational cost, and overall performance. For instance, how does the choice of segmentation model (e.g., SAM vs. SAM2) affect performance when considering both accuracy and resource consumption? Understanding these trade-offs could help practitioners decide how best to apply PSP in environments with varying levels of available resources.
> > > >
> > > > 3. **Resource-Constrained Scenarios**: While you've provided insights into the computational overhead during training, it would be helpful to understand more about how these factors might impact real-world applications, particularly in resource-constrained environments such as robotics or embedded systems. For instance, could there be strategies for optimizing or simplifying PSP to make it more feasible in such contexts?
> > > >
> > > > These points are intended to provide additional clarity on the impact of your approach in a broader range of settings and could significantly enhance the practical utility of your work. Again, I appreciate the depth of your response and the additional experiments conducted. My evaluation is intended to reflect both the strengths of your contribution and the areas where further refinement could make the work even more impactful.

---

### Official Review · Reviewer_7m6w · 2024-07-08

**Soundness:** 2
**Presentation:** 2
**Contribution:** 2
**Rating:** 4
**Confidence:** 4

**Summary:**

This paper presents a novel model-based reinforcement learning (MBRL) method that focuses on important parts of image-based environments with distractions, aiming to improve policy learning. The proposed method introduces gradient-based weighting, segmentation-based aggregation, and adversarial action prediction to world models. Experiments are conducted using the DeepMind Control Suite environment with varying levels of distractions.

**Strengths:**

1. The paper introduces a new method that utilizes policy-gradient weighting and object-based weighting for image reconstruction.
2. It devises a Reafferent DeepMind Control environment where the distracting background deterministically depends on the agent’s previous action and the elapsed time within an episode.

**Weaknesses:**

1. In Figure 1, is the arrow between the $z_i$ and the image decoder reversed? The notation of $i$ is confusing, as it is unclear whether it represents a time step or an image pixel (Line 86). The arrows between $z_i$ and $v_i$, $a_i$ seem to suggest that a value model and a policy model are learned in the world model learning phase. However, in my opinion, these two models should be learned in the downstream RL phase.
2. I wonder if the policy used to learn policy-gradient weight is the downstream policy that aims to maximize value. If I understand correctly, the policy model is used, but its weights are not updated in the world model learning phase. Additionally, how the connection between the latent state $s$ and the image pixel in Eq (1) is established, considering that the input of the policy model is based on $s$.
3. I have some doubts about the Segment Anything Model (SAM). Was it fine-tuned in the experiment? It would be helpful if the authors could provide more details on how SAM is used in this work and what the output of $SEG(x_i)$ is. SAM outputs multiple masks for different objects based on prompts, so how do you determine which mask to use?
4. In Eq (4), inferring an action from the state of a single time step seems unreasonable. In general, actions are inferred from successive states. Moreover, I do not agree with the statement that the encoder extracting information about the agent's action is wasteful. A more reasonable explanation is needed.
5. It is strongly encouraged for the authors to provide visual results of $W$ and segmentation masks in the experiments to facilitate clearer explanations and analysis.
6. The experimental environment and tasks are too limited and simplistic to demonstrate the effectiveness of the proposed method. It would be beneficial to use a more realistic environment, such as CARLA, which also includes various distractions related and unrelated to the agent's actions.
7. The performance of the proposed method appears relatively poor based on Figure 6, A1, and Tables 1 and 2. Specifically, it is challenging to conclude from Table 1 that the proposed components improve the model's performance.
8. In Figure 5, the comparison images are not from the same point in time or the same episode, which does not support the stated conclusion. Additionally, for a more accurate performance comparison, it would be more appropriate to provide an MSE result that only computes the agent region.

**Questions:**

Please see the weaknesses section.

**Limitations:**

Limitations are discussed in the paper.

---

> ### Author Rebuttal · Authors · 2024-08-07
>
> We thank the reviewer for their detailed and helpful comments.
>
> **1.** We are grateful to the reviewer for catching this typo. In Figure 1, we should be using *t* as the subscript, not *i*. We will update the figure and text accordingly.  We will additionally make clear that while v_i and a_i are learned in a downstream RL phase, they depend on the latent state z_i.
>
> **2.** Thank you for the opportunity to clarify. The policy used to weight reconstruction loss is the same policy trained downstream to maximize value. Model training in the Dreamer architecture consists of alternating phases of world model updating (based on action sequences in the replay buffer) and policy/value learning (using imagined action sequences that leverage the world model). The policy is dependent on the latent state, and the latent state is dependent on the image. The reviewer’s understanding is correct that neither the policy nor the value model weights are changed during the world model learning phase. However, the policy depends on the latent state, and the latent state depends on the image pixels. The gradient between the policy and the image pixel is established via standard auto-differentiation (with gradients that flow through the latent state). We will update the text to clarify, thank you.
>
> **3.** Thank you for the opportunity to clarify details about our use of SAM. In our experiments, SAM is not finetuned. For our implementation, in order to determine our segmentation masks, SAM is prompted with a grid of 256 points. The resulting masks are filtered via metrics computed by the SAM algorithm, specifically a prediction IOU (intersection over union) thresh and a “stability score", followed by non-maximum suppression (NMS). Finally, pixels that have not been assigned to a segmented object are grouped into a single extra object, to ensure that they are not ignored entirely. The outcome of this procedure is a segment assignment for each pixel. We will update the text to clarify this procedure.
>
> **4.** Thank you for providing us the opportunity to further clarify the role of the action-prediction head specified in Eq. 4. The intuition is that the current state *may* contain information about the preceding action. You are completely correct that in general, a past action cannot be inferred from just a single timestep. However, in *some* cases they can. Our assertion is that maintaining this information in the image embedding is redundant and a waste of capacity, *since the the previous action is already provided as input to the world model*. This can become a particularly problematic source of wasted capacity when extensive or detailed aspects of the image are related to the previous action, and thus are redundant. The action prediction head allows us to remove this redundant information from the image embedding step.
>
> We found that this intuition was supported by our empirical results. Notably, realize we failed to include results from a key ablation that we had run: policy gradient weighting + segmentation but without the action head. We have now added it (Fig. R1I, see the second row), and it makes it clear the performance improvements from including the action head (the first row), in both the unaltered and reafferent environments.
>
> **5.** Thank you for this great suggestion, we will provide additional examples of segmentations and weighting masks , such as Figs. R1B,F,G.
>
> **6.** We appreciate the reviewer's interest in application to more complex settings. We chose the current settings because they allowed for direct comparison across a wide variety of techniques that had been tuned for Deepmind Control Suite and related environments, and provided a good foundation for our new Reafferent environment. Extensions to environments such as CARLA are a great direction for future work that we are excited to pursue.
>
> **7.** Our claim based on the experimental results is that PSP yields: an unambiguously better performance on the most challenging distractors (e.g. Reafferent), a more consistently good result on the Distracting Control Suite (it is in second place on both tasks, while the first place method is different in each case), and adequate performance that is the same or better than baseline DreamerV3 on the unmodified Control Suite. The data in Figure 3, 6, A1 and Table 1 support these claims. A similar conclusion comes from Table 2, in which the combination of methods yields good performance on both the unaltered and reafferent environments. While we see that it is possible to achieve even better performance on Reafferent environment, this can come at the cost of dramatically reducing performance in non-distracting settings (as is the case with value-gradient or policy-gradient weighting alone, in which there is a stark tradeoff in performance between the distracting and non-distracting settings), and the full PSP model therefore performs better than the ablations across settings.
>
> **8.** Thank you for this suggestion. We have now included a comparison image at the time point in the same episode that clearly exhibits the superior model performance of the PSP agent on the Reafferent environment (Figure R1H).

---

> > ### Comment · Reviewer_7m6w · 2024-08-13
> >
> > Thank you for your detailed response. I have adjusted the score to 4 due to lingering doubts regarding the effectiveness of the proposed method.

---

> > > ### Author Response · Authors · 2024-08-13
> > > **Reply**
> > >
> > > We are very grateful for the reviewer's time and engagement with our responses. Thank you.

---

### Official Review · Reviewer_8n3B · 2024-07-12

**Soundness:** 2
**Presentation:** 2
**Contribution:** 3
**Rating:** 5
**Confidence:** 4

**Summary:**

### Review Summary

This paper presents a novel approach to improving model-based reinforcement learning (MBRL) by identifying that detailed but irrelevant aspects of the world can exhaust the capacity of the world model, thus hindering the learning of important environment dynamics. The proposed method, Policy-Shaped Prediction (PSP), uses a pretrained segmentation model, a task-aware reconstruction loss, and adversarial learning to focus the world model's capacity on relevant environment features. The paper demonstrates that PSP outperforms other approaches, including DreamerV3 and DreamerPro, in environments with intricate but irrelevant background distractions.

### Major Concerns

1. **Effectiveness of Policy Gradient Information**: The utility of the policy gradient information when the policy is not sufficiently good is questionable. This raises a chicken-and-egg problem: if the policy is poor, the gradient information might not effectively guide the model's learning, but if the policy is already good, further gradient information might not be necessary. Clarifying this point is crucial to support the core contribution of the paper. I have the following guess given the positive results: either the gradient information provided early on is sufficiently informative or the policy and model improve together, even if initial gradients are inaccurate. The authors should provide further explanation on this aspect.

2. **Motivation Comparison with Related Work**: The motivation emphasized in the paper—to distinguish between decision-relevant foreground and irrelevant background during model learning—has been similarly addressed in previous works on Block MDPs and invariant representations (e.g., [1], [2]). These works, although employing different technical approaches, share a similar motivation. The authors should compare their approach with these works, providing theoretical or experimental analysis to highlight the differences and advantages.

3. **Dependence on Pretrained Segmentation Model**: Given that the segmentation model is pretrained, what is its range of applicability? How much do the final results depend on the effectiveness of the segmentation model? The authors only conducted experiments in two environments and did not explore this dependency, raising questions about the generalizability of the approach to new environments.

4. **Experimental Section Clarity**: The experimental section starts with good questions and appears to be structured to answer them. However, the current presentation makes it difficult for readers to find the answers and understand which experiments address which questions. Revising the presentation for clarity would help make the experimental logic more apparent.

5. **Applicability to Different Scenarios**: The three methods proposed (policy gradient with respect to state, segmentation, adversarial training loss) seem tailored to specific problems (e.g., decision depend on object segmentation, clear object boundaries). If the problem scenario changes slightly, will these methods still work? Even considering the current scenario, how many common RL problems can this approach cover? The experiments are primarily conducted on two tasks, which weakens the persuasiveness of the results.

### References

[1] Learning Invariant Representations for Reinforcement Learning without Reconstruction
[2] Invariant Causal Prediction for Block MDPs

**Strengths:**

See above

**Weaknesses:**

See above

**Questions:**

See above

**Limitations:**

See above

---

> ### Author Rebuttal · Authors · 2024-08-06
>
> We thank the reviewer for their insightful comments.
>
> **Effectiveness of Policy Gradient Information**: We do see some evidence of the chicken-and-egg problem. Importantly, though, our interpolated weighting term (see Eq. 3 in the manuscript) provides a route for escaping this problem by allowing the model and policy to always have access to cues from the entire state and to improve together. Thus, even if the initial policy-weighting is bad the model still has the chance to globally improve its predictions and recover from the initially bad weighting. As might be expected, following your great intuition, we find that the interpolation specified by Eq. 3 was essential to achieving consistently good performance.
> To demonstrate the value of this interpolation step, we ran an experiment comparing agents with and without the interpolation, on Cheetah Run with Reafferent distraction. As shown in Fig. R1C, without interpolation (green line), the agent is stuck with poor performance for nearly 150K steps, until it eventually does manage to escape the chicken-and-egg problem. However, when including interpolation (blue line) we see that the agent is able to quickly achieve nonzero performance and begin steadily improving.
> Additionally, in Fig. R1F, we demonstrate how the salience map improves throughout training, suggesting that the policy and world model can improve together with training.
>
> **Motivation Comparison with Related Work**: Thank you for bringing up the interesting connection to invariant representations. Indeed these methods are quite relevant, and it is valuable to consider methods for identifying state abstractions that leverage bisimulation [1] or the closely related concept of invariant causal feature prediction [2]. The DBC algorithm from [1] is most relevant to our context, as it is designed explicitly for high dimensional RL applications with distractions. On the other hand, the MISA algorithm from [2] requires multiple distinctly labeled experimental settings, and while they demonstrate imitation learning from high dimensional images, extension to high dimensional RL applications is not necessarily straightforward and could be a novel contribution in its own right.
>
> However, we do not think that it is apt to experimentally test DBC as a baseline in our setting for two key reasons: 1) The Denoised MDP work [Wang 2023] compares directly against DBC and demonstrates substantially better performance than DBC (see Table 1 in Wang 2023). In turn, our experiments compared directly against Denoised MDP and we observed that PSP exhibited substantially better performance than Denoised MDP. 2) Our focus is on model-based RL, but only a model-free version of DBC was published and it is not straightforward to consider how it would be applied in the context of Dreamer-type architectures.  For these reasons, we believe that our current experimental baselines are adequate to encompass these approaches. However, we think the connection is important and we will add discussion of invariant representations to the manuscript. Thank you!
>
> **Dependence on Pretrained Segmentation Model**:
> We acknowledge that the use of a pretrained segmentation model has the potential to limit the range of applicability, however we also believe that foundational segmentation models such as SAM have grown robust enough to be considered for these applications. Moreover, future work will improve these models, with cheaper inference and higher quality and more generalizable segmentation – as evidenced by the recent release of SAM2.
> We have now included experiments with an *additional* segmentation model, the 'tiny' variant of SAM2, and observe similarly good results as with SAM (Fig. R1A).
> Additionally, we note the segmentation model does *not* have to be perfect. The SAM segmentation often splits apart what should be considered a single object. In Fig. R1B, for example, the cheetah is split into two, three, or four segments throughout a successful run. This suggests a degree of insensitivity to exact details of the segmentation.
> Notably, the Deepmind Control Suite setting is likely out of the domain of the data on which SAM or SAM2 were trained, and yet segmentations from these algorithms are still adequate to achieve good PSP performance.
>
> **Experimental Section Clarity**: Thank you. We will clarify the results section to explicitly indicate how we answer each of our listed experimental question (e.g. by adding references to Q1, Q2, etc. in the appropriate locations). Specifically: Q1 is answered in lines 199-228, Figure 3, and Table 1. Q2 is addressed by Figure 5 (and additional supplemental images of salience maps that we will include). Q3 is addressed by Figure 6 and Table 1 at line 233. Q4 is addressed by line 230, Figure A1, and Table 1. Q5 is addressed at lines 240-255, and Table 2.
>
> **Applicability to Different Scenarios**: We selected Deepmind Control Suite as the base environment to allow for direct comparison against a number of existing baselines, in the settings that they were demonstrated. Distracting Control Suite was the only environment shared across the DreamerPro, DenoisedMDPs, and Task Informed Abstractions baselines. Furthermore, we additionally incorporate the Reafferent environment to extend beyond Distracting Control Suite into more challenging scenarios. We note that the segmentation-based approach of PSP does assume object-based settings, however this assumption is relevant to a wide variety of problems. We acknowledge that there may be limitations of our approach in certain other scenarios, although it is not clear a priori what those limitations may be, and we look forward to pursuing future work to investigate this.

---

> > ### Comment · Reviewer_8n3B · 2024-08-13
> >
> > Thank you to the author for the detailed response. Overall speaking, the author has addressed some of my concerns, so I have raised my score to 5. My remaining major concern is about the generalizability of the proposed method, which involves two aspects:
> >
> > 1. If the key to the decision-making problem is not largely determined by object segmentation, is the proposed method still effective?
> > 2. The generalizability of the Pretrained Segmentation Model—specifically, in what range can it still work if applied to non-natural images?

---

> > > ### Author Response · Authors · 2024-08-13
> > > **Reply**
> > >
> > > We are very appreciative of the reviewer's time and engagement with our responses. Thank you.
> > >
> > > While it is true that this may not apply broadly to decision-making scenarios that do not depend on object segmentation, there are many important scenarios that do (especially in the visual domain). Moreover, it is conceivable that our approach could still apply in those settings, with the segmentation simply serving as a reasonable method for reducing noise in the gradient-based weighting. We hope to investigate this in future work.
> > >
> > > In terms of the generalizability of the segmentation model, we think that our existing evidence is actually quite corroborative of good generalization to non-natural images. SAM and SAM2, it turns out, are quite effective (i.e. effective enough to work well with PSP) in the very non-natural Deepmind Control Suite and our even less-natural distracting variants. It is possible other settings may present more of a challenge, though we also expect these segmentation models to improve as the datasets used to train them continue to grow. Thank you again for your comments and insight, we really appreciate the discussion.

---

### Author Rebuttal · Authors · 2024-08-05

### Overview for all reviewers
We thank the reviewers for their excellent thoughtful and constructive comments. We agree with the reviewers' assessment that Policy-Shaped Prediction (PSP) is a novel and effective method to address a well-motivated and relevant problem: reducing the influence of distracting information in model-based RL. We will address each reviewer with individual responses.

We want to highlight that we have now extended PSP to use an **additional segmentation algorithm** and demonstrated ***similar performance but with substantially reduced resource consumption***. All four reviewers raised questions regarding the segmentation algorithm that we used (SAM), including the generalizability, required accuracy, and resource requirements of SAM. We believe we have addressed these questions with our new experiments.

Our initial selection of SAM was guided by the belief that it represented just the first of many segmentation algorithms that are sure to follow with improved generalizability, performance, and resource consumption. This belief was corroborated by the recent release of SAM2 (on July 29), which according to the associated manuscript (Ravi, et al. 2024) exhibits both improved performance *and* as little as 17% the required resources (e.g. 6x the segmentation speed). We updated PSP to also be able to use SAM2, using the 'tiny' model, the smallest and lowest accuracy of the provided model sizes. Our basic implementation immediately reduced resource consumption by 2x, while yielding a score on Reafferent Cheetah of 360.2, within the range of our original results with SAM of 383.1 ± 23.8 (note: we only had time/resources available to run one seed, but we would include additional seeds in a final manuscript). Optimizations, including using SAM2's video segmentation capabilities and better utilization of the GPU, would likely further improve this resource consumption. These results also suggest that our implementation of PSP is *not* highly sensitive to the segmentation algorithm.

In the attached pdf (Figure R1), we have also included additional ablations that highlight the importance of the mask interpolation step (Eq. 3) and the action prediction head (Eq. 4), and more visualizations of segmentation masks and salience maps.

**Together, these results bolster the claim that PSP is an effective method for reducing the impact of challenging distractors**, and that it also has the potential to benefit from gains in related realms such as generalizable segmentation.

---

### Comment · Area_Chair_ppru · 2024-08-13
**TLDR: Reviewers please do acknowledge the rebuttals and react to them.**

Dear reviewers,

thanks for your reviews.  Please do look at the rebuttals of the author and give the authors some feedback whether they could address your concerns.  This is really important for the authors.

Best regards Your AC.

---

### Decision · Program_Chairs · 2024-09-25

**Decision:**

Accept (poster)

**Comment:**

Scores 6 5 4 5

The paper proposes a new approach to avoid spending learning capacity on predictable features of the world that are irrelevant to the agent's decisions.  The initial concerns of the reviewers could be alleviated by the rebuttals also with new meaningful experiments.  A second concern is the added complexity for which the reviewers would like to have a more analytical analysis.  That would be definitely a useful addition and it would improve the paper.  Nonetheless, the reviews are quite positive about this work that has a good contribution and is well written and shows relevant empirical results with lots of details for reproducibility.  Thus I suggest to accept this paper.